# 3D-MINFLUX nanoscopy reveals distinct allosteric mechanisms for activation and modulation of PIEZO1 by Yoda1

Clement Verkest ⓘ , Lucas Roettger ⓘ , Nadja Zeitzschel ⓘ &
Stefan G. Lechner ⓘ ✉

The small molecule Yoda1 has become an indispensable tool for dissecting the role of the mechanically activated ion channel PIEZO1 in physiological and pathological contexts. Previous studies proposed that Yoda1 first binds to a hydrophobic pocket at the interface of the transmembrane helical units 8 and 9 close to the inner leaflet of the membrane to induce conformational flattening and thus channel activation and then transitions to a deeper located binding site with higher affinity in the open conformation. Here, using site-directed mutagenesis, electrophysiology, computational modelling and 3D-MINFLUX nanoscopy, we refine this model by demonstrating that mutation of the previously proposed Yoda1 binding site-I, solely abolishes Yoda1-induced activation and channel flattening but preserves modulation of mechanically-evoked PIEZO1 currents, whereas mutation of F1715, which lines a transient binding cavity accessible only in the flattened PIEZO1 conformation, eliminated modulation without affecting Yoda-induced calcium entry. Thus, our data support a revised model for Yoda's mode of action that distinguishes discrete allosteric pathways for PIEZO1 activation versus modulation and provides a framework for the design of next-generation use-dependent PIEZO1 modulators.

Mechanotransduction underpins a vast array of physiological processes by which cells sense and respond to mechanical stimuli, translating physical forces into biochemical signals that direct cell behaviour. A key player in this process is PIEZO1, a large, trimeric, mechanically-gated ion channel that converts membrane tension and deformation as well as shear forces into ionic fluxes, thereby initiating downstream signalling cascades[1–6]. PIEZO1 has been recognized as a ubiquitous sensor of mechanical cues across tissues with its role extending from embryonic development to adult homeostasis, where it orchestrates processes as diverse as vascular remodelling, erythrocyte volume regulation, lymphatic valve formation, baroreception, cell division and cell migration[7–11].

Cryogenic electron microscopy (cryo-EM) has elucidated the architecture of PIEZO1, revealing a trimeric, propeller-shaped complex with each protomer comprising 38 transmembrane helices organized into discrete structural modules: peripheral blade domains composed of nine transmembrane helical units (THU1–9)−each comprising 4 transmembrane α-helices, an extracellular cap, intracellular beam helices acting as lever arms, and a central ion-conducting pore module[12–14]. The blade domains emerge from the central pore at an acute angle, arching upward and outward to form a curved dome within the membrane. Under mechanical tension, this dome is proposed to flatten, transmitting force through the long intracellular beam helices to the pore module[15–17]. The beam domains connect the distal blades to the central pore's lower vestibule and are thought to amplify conformational changes, acting as rigid levers that convert subtle blade movements into pore opening[18–20]. This unique modular architecture enables PIEZO1 to directly sense membrane tension and undergo conformational transitions that open the central pore in response to mechanical stimuli.

Department of Anaesthesiology, University Medical Center Hamburg-Eppendorf, Hamburg, Germany. ✉e-mail: s.lechner@uke.de

Traditionally, the investigation of PIEZO1 function has relied heavily on patch-clamp assays such as the cell-attached pressure-clamp and the whole-cell poking technique, which remain the gold standard for characterizing mechanosensitive currents[21,22]. These techniques are, however, labour-intensive and low throughput, motivating the search for pharmacological tools that can modulate PIEZO1 activity independent of direct mechanical manipulation. The discovery of Yoda1, a small molecule agonist of PIEZO1[23], has deeply impacted the experimental landscape by providing a means to chemically activate and modulate the channel in cellular assays, such that Yoda1 has become an indispensable and widely used tool for dissecting the roles of PIEZO1-dependent mechanotransduction in physiological and pathological contexts[24]. However, the reliance on Yoda1 as a surrogate activator necessitates a deep mechanistic understanding of its mode of action to ensure that observed effects truly reflect changes in mechanosensitivity rather than artifactual Yoda-induced calcium overload.

The currently prevailing view about Yoda's mode of action is largely based on a series of seminal studies from the Lacroix and Luo labs, who have combined molecular dynamics simulations with site-directed mutagenesis and electrophysiology to elucidate the mechanism underlying Yoda1-induced activation of PIEZO1. They identified the interface between THU8 and THU9 as a crucial region for Yoda1-dependent activation of PIEZO1[25] and proposed that Yoda1 acts as a molecular wedge that binds to a hydrophobic pocket lined by residues A2091, A2094, and A1718[26] thereby driving the channel into a flattened state analogous to that induced by membrane tension. This allosteric effect supposedly propagates through the beam helices, lowering the energy barrier for pore opening and prolonging the open state by slowing inactivation[18]. More recently, Jiang and colleagues further refined this model using molecular dynamics–based absolute binding free energy simulations, which suggested that Yoda1 may transition to another amphipathic binding site in PIEZO1's open state, which has a higher affinity and is located deeper within the THU8–9 interface[27]. Using in-silico docking of a virtual compound library onto this deeper binding site in their PIEZO1 open state model, Jiang et al. could identify two novel PIEZO1 activators, which strongly supports a two-site induced-fit-like mechanism of action of Yoda1. Using MIN-FLUX super-resolution nanoscopy of PIEZO1 in fully intact cells, we and others have recently corroborated the basic hypothesis that Yoda1 induces PIEZO1 flattening, by showing that the distance between the distal ends of the blades, which is a measure for the degree of flattening, indeed increases in the presence of Yoda1[28,29].

In this work, we set out to utilize 3D-MINFLUX nanoscopy, which had previously enabled us to resolve subtle conformational changes in PIEZO1 in intact cells[29], to decipher the nature and choreography of atomic interactions that underlie Yoda-binding and Yoda-induced conformational changes in PIEZO1.

## Results

### Disruption of the putative allosteric binding site abolishes Yoda1-induced PIEZO1 activation but not modulation

Previous studies proposed that Yoda1 binds to a pocket in the THU8–9 interface lined by A2091, A2094, and A1718 in the curved state (Fig. 1A)[26] and transitions to a deeper located binding site with a higher affinity as the channel conformation changes to the flat state[27]. Specifically, Botello-Smith and colleagues showed that replacing A2091 with tryptophan (A2091W) partially reduced Yoda1-induced calcium entry, whereas A2094W and A1718W almost completely abolished it. However, A1718W was also mechanically insensitive, preventing conclusions about its specific role in Yoda1 binding. In contrast, A2094W responded normally to mechanical stimuli, implicating A2094 specifically in chemical activation by Yoda1.

To investigate how Yoda1 induces flattening at the single-molecule level, we reproduced the PIEZO1.A2094W mutant and first characterized it by GCaMP8 calcium imaging and patch-clamp electrophysiology in Neuro2a PIEZO1 knockout cells (N2a-P1KO)[30]. As reported[26], even supramaximal Yoda1 concentrations (10 and 100 μM) failed to elicit calcium influx in A2094W-expressing cells (Fig. 1B), whereas mechanically-evoked A2094W-currents were indistinguishable from PIEZO1 wildtype currents in the pressure-clamp and poking assays (compare CTL in Fig. 1C, D), respectively, indicating that the machinery and the intramolecular force-transmission pathways required for detecting and converting mechanical stimuli into electrical signals were not affected by the tryptophane substitution. Surprisingly, however, Yoda1 robustly sensitized and modulated mechanically-evoked PIEZO1.A2094W currents to the same extent as wildtype PIEZO1 currents (compare Fig. 1C, D). Thus, Yoda1 significantly slowed down the inactivation kinetics of pressure-evoked PIEZO1 and PIEZO1.A2094W currents and reduced the half-maximal pressure (P50) required for their activation (Fig. 1C). Likewise, both poking-evoked PIEZO1 and PIEZO1.A2094W currents in whole-cell recordings exhibited slower inactivation, lower activation thresholds and larger amplitudes in the presence of Yoda1 (Fig. 1D).

Together, these data challenge the presumed role of A2094 in Yoda1 binding and indicate that chemical activation and modulation of mechanically-evoked PIEZO1 currents may involve distinct molecular mechanisms. Two non-exclusive explanations are: (i) A2094 is dispensable for Yoda1 binding but critical for allosterically coupling Yoda1-induced conformational changes to pore opening, or (ii) the A2094-containing binding site is required for channel activation, whereas a separate site mediates Yoda1-dependent modulation of mechanically-evoked currents.

### Computational modelling reveals an alternative Yoda1 binding site that becomes accessible in the flattened state

Regarding the possible existence of alternative Yoda1 binding sites, an intriguing possibility was recently raised by Jiang and colleagues, who performed molecular dynamics–based absolute binding free energy simulations, which suggested that Yoda1 may bind to an alternative amphipathic binding site in PIEZO1's open state, which has a higher affinity and is located deeper within the THU8–9 interface[27]. This hypothesis is strongly supported by the discovery of two PIEZO1 activators that were found by virtually docking a compound library onto this deeper located binding site. Another stable binding site, which was identified by the molecular dynamics simulation that originally led to the discovery of A2094[26], may be present at the interface between the beam domain and THU9 (Fig. 1A). Botello-Smith and colleagues[26], however, ruled out a contribution of this site, because point mutations did not abolish Yoda1-dependent activation of PIEZO1. Considering that activation and modulation of PIEZO1 by Yoda1 might, however, be mediated by different mechanism, we hypothesized that this alternative binding site might be exclusively required for modulation. To test this hypothesis, we substituted the PIEZO1 beam residues that are located at the beam–THU9 interface with the corresponding amino acids of PIEZO2, which is not sensitive to Yoda1 (P1.S1330_N1332delinsA1479_K1481[PIEZO2]). Although Yoda1 failed to modulate poking-evoked currents mediated by this mutant, Yoda1-evoked Ca²⁺ influx in GCamp8 imaging was normal and stretch-evoked currents exhibited slowed inactivation and reduced P50 values in the presence of Yoda1 (Supplementary Fig. 1A–D), indicating that the beam–THU9 interface is not the principal Yoda1 binding site. Hence, the selective loss of modulation of poking-evoked currents in this mutant, most likely results from compromised allosteric coupling, which is consistent with previous reports demonstrating an important role of the beam domain in relaying Yoda1- and poking-induced conformational changes in the channel periphery to the central pore domain, possibly by a lever-like mechanism[14,18,31].

We thus next considered the deeper located binding site proposed by Jiang et al., which becomes available in the open state. Jiang

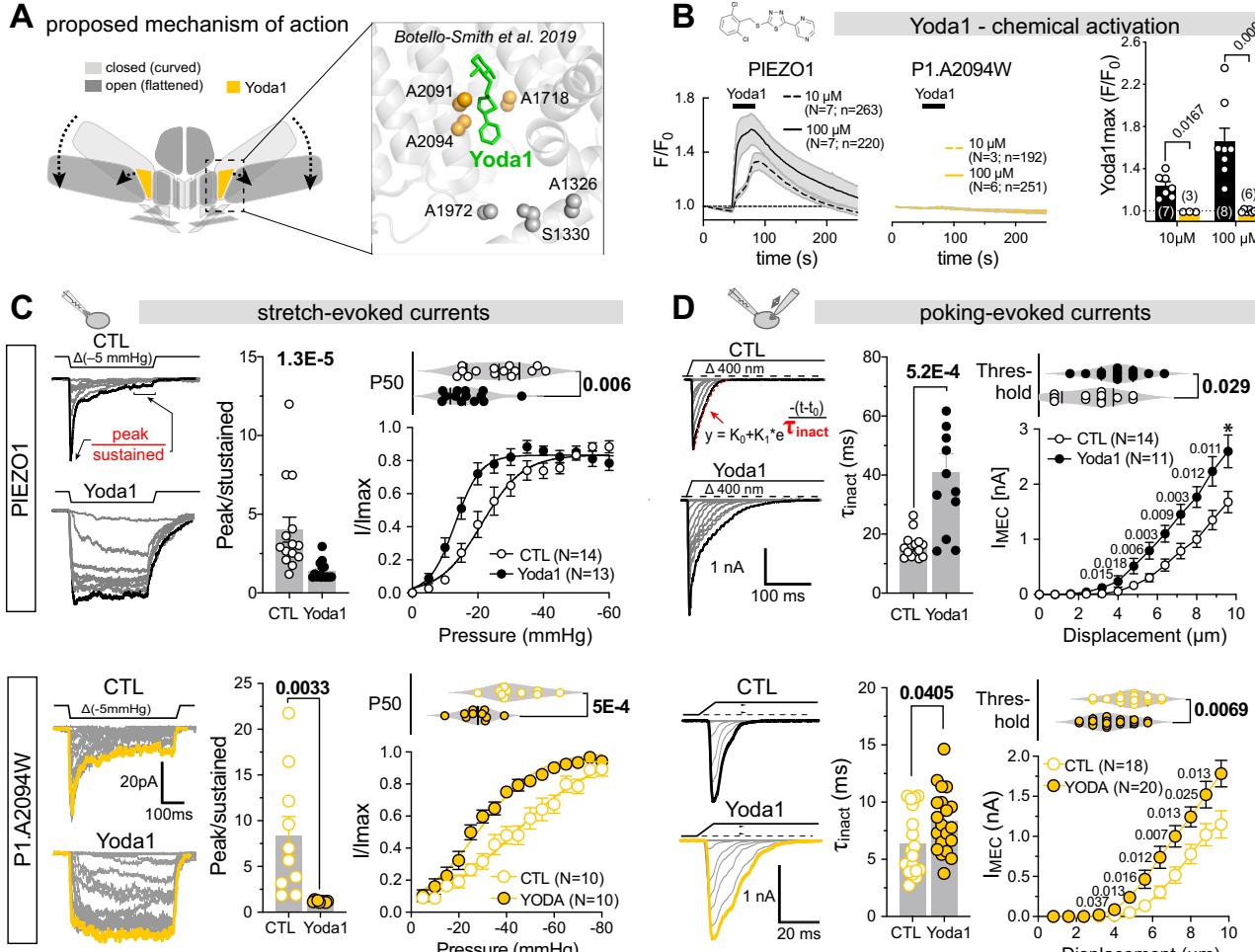

**Fig. 1 | Mutation of the proposed Yoda1 binding site abolishes activation but not modulation of PIEZO1. A** Proposed mechanism of action of Yoda1 on PIEZO1 and close-up view of the previously proposed Yoda1 binding sites in sphere representation. **B** time courses of mean ± SEM Ca²⁺-influx (F/F₀, GCamp8) evoked by 10 and 100 μM Yoda1 in cells expressing PIEZO1 (left) and P1-A2094W (center) and comparison (two-sided Mann–Whitney test) of the mean ± SEM maximum responses (right). "*N*" denotes the numbers of coverslips and "*n*" denotes the total number of cells. Bars (right panel) represent means ± SEM and circles show the mean population responses of individual coverslips (*N*-numbers provide in bars). **C** Modulation of PIEZO1 (top) and P1.A2094W (bottom) stretch-evoked currents by 30 μM Yoda1. Example traces evoked by incrementing pressure stimuli (left), comparison of peak/sustained ratio (middle) using two-sided Mann–Whitney test (PIEZO1: CTL = 4.02, *N* = 14 vs Yoda1 1.38, *N* = 15, *P* = 0.0000131) and Student's *t*-test (P1.A2094W: CTL = 8.35, *N* = 10 vs Yoda1 1.19, *N* = 10, *P* = 0.0033), normalized pressure-response curves (bottom right) and comparison of P₅₀ values with and without Yoda1 using two-sided Student's *t*-test (PIEZO1: CTL = 26.1 ± 8.9 mmHg,

*N* = 14 vs Yoda1 = −16.9 ± 6.65 mmHg, *N* = 13, *P* = 0.006) and Mann–Whitney test (P1.A2094W: CTL = −43.7 ± 10.04 mmHg, *N* = 10 vs Yoda1 = −27.4 ± 6.7 mmHg, *N* = 10, *P* = 0.0005). **D** Modulation of PIEZO1 (top) and P1.A2094W (bottom) poking-evoked currents by 30 μM Yoda1. Example traces evoked by incrementing poking stimuli (Δ 400 nm, left), comparison of inactivation time constants obtained with exponential decay fit (middle) using two-sided Mann–Whitney test (PIEZO1: CTL = 15.9 ± 4.3 ms, *N* = 14 vs Yoda1 = 41.1 ± 21.6, *N* = 12, *P* = 0.000258) and Student's *t*-test (P1.A2094W: CTL = 6.39 ± 2.8 ms, *N* = 18 vs Yoda1 = 8.3 ± 2.7, *N* = 20, *P* = 0.0405), displacement-response curves (bottom, right) and comparison of mechanical activation thresholds using two-sided Mann–Whitney test (PIEZO1: CTL = 4.1 ± 1.2 μm, *N* = 14 vs Yoda1 = 2.8 ± 1.4 μm, *N* = 12, *P* = 0.029; P1.A2094W: CTL = 4.8 ± 1.04 μm, *N* = 18 vs Yoda1 = 3.8 ± 1.1 μm, *N* = 20, *P* = 0.0069). *N*-numbers in (**C**, **D**) refer to biological replicates (number of recorded cells). Response amplitudes at different displacements were compared using two-sided Mann–Whitney test without adjustments for multiple comparison and *P*-values are provided next to symbols in graphs.

and colleagues had identified this binding site using an in-silico modelled open structure, because at the time the study was published cryo-EM data for the flattened PIEZO1 state was not available. Meanwhile, high-resolution cryo-EM structures of an intermediate and a fully flattened state have been resolved[15,16], which revealed substantial rearrangements in the chemical environment of the supposed Yoda binding pocket, suggesting that alternative binding pockets may indeed emerge upon channel flattening. To test this hypothesis, we analysed the cryo-EM structures of the curved (PDB:7WLT) and flattened (PDB:7WLU) states of PIEZO1[15,16] using the DoGSite3 algorithm, a grid-based method that uses a Difference of Gaussian filter to detect potential ligand binding pockets[32]. Consistent with the hypothesis put forward by Jiang et al., this analysis identified different putative

binding pockets in the curved and flat conformation. Thus, in the curved conformation one pocket was found subjacent to the inner leaflet of the membrane at the interface between the beam and THU8/9 (pocket 1, Fig. 2A) and another one at the interface between THU8 and THU9 within the plasma membrane (pocket 2, Fig. 2A). In the flattened state, pocket 1 at the beam–THU8/9 interface appears to be relocated counterclockwise (when viewed from above) and further away from the beam, while pocket 2 remains roughly in the same position. Most importantly, in the flattened state a third prominent pocket emerged that is buried deep within the THU8–9 interface and adjacent to pocket 2 (orange mesh in Fig. 2A). To independently test if Yoda1 could potentially bind to these cavities, we next calculated possible docking poses of Yoda1 using the AutoDock Vina[33,34]. Using

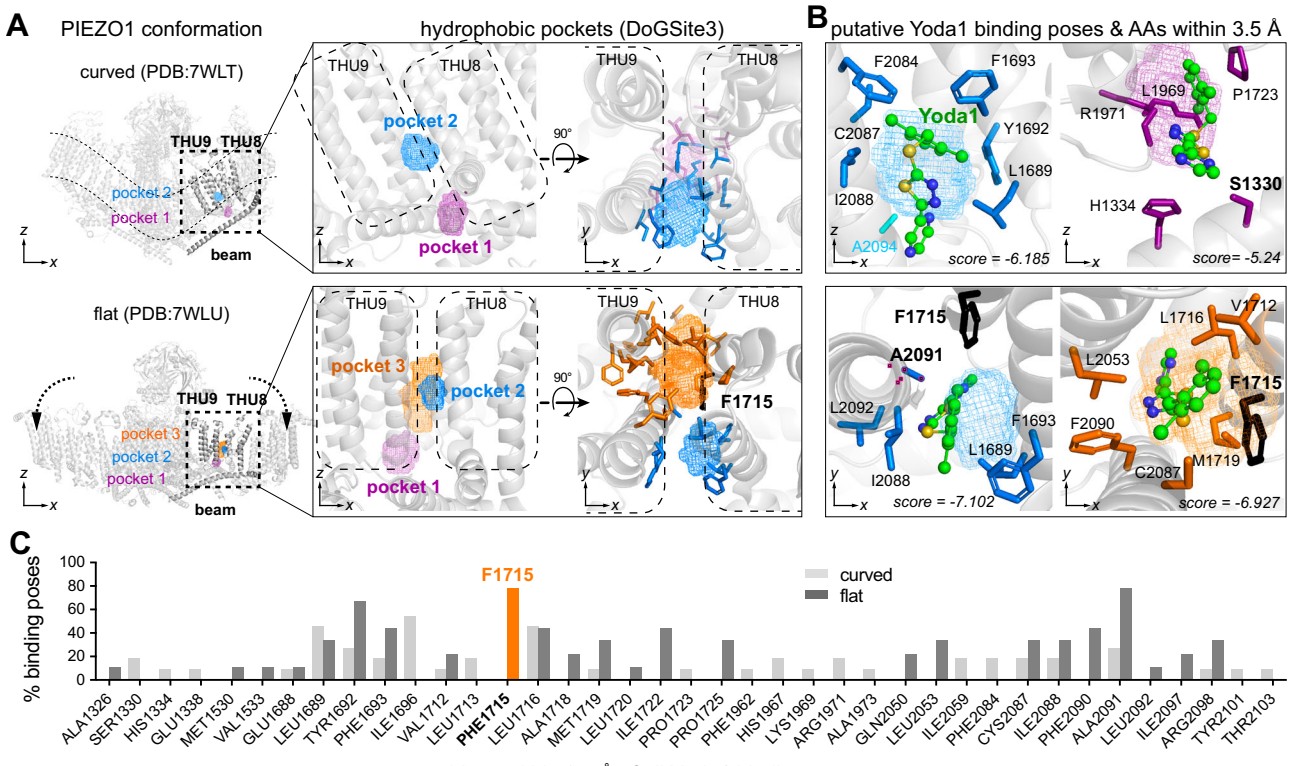

**Fig. 2 | PIEZO1 flattening exposes putative alternative Yoda1 binding sites lined by F1715. A** Side view of curved (PDB:7WTL, top left) and flattened (PDB:7WTU, bottom left) PIEZO1 cryo-EM structure with close-up side (middle) and top (right) view of transmembrane helical units (THU) 8–9 interfaces with the putative binding pockets detected by DoGSite3 algorithm shown in isomesh representation. Amino acid side chains that line the pockets are shown in stick representation. **B** Close-up views of Yoda1 binding poses detected by AutoDock Vina in the curved (top) and flattened (bottom) PIEZO1 conformation. Note, only the binding poses with the highest scores that align with the binding pockets detected in (**A**) are shown. Residues that are within a distance of 3.5 Å of Yoda1 are shown in stick representation. Note, F1715 appears to be involved in Yoda1 binding in pocket 2 and pocket 3 in the flat conformation. **C** AutoDock Vina detects multiple possible docking poses for Yoda1. The bar graph shows the percentage of docking poses in which the indicated amino acids are located within 3.5 Å of Yoda1. Note, F1715 is in close proximity of Yoda1 in 78% of all possible binding poses indicating an important role of F1715 in Yoda1 binding in the flat state.

this approach, we identified 9 putative binding poses in pocket 2 at the THU8–9 interface and two poses in pocket 1 in the curved conformation (poses with highest scores are shown in Fig. 2B). Notably, in the flattened conformation, no binding poses were found in pocket 1, but 5 poses in pocket 2 and 4 poses in pocket 3, which only emerges in the flattened state (Fig. 2B). Note, AutoDock Vina also identified binding poses that did not align with the pockets detected by DoGSite3, but these poses were distant from the THU8–9 interface and should be interpreted with caution, because we used incomplete structural models that only contained THU8, THU9 and the subjacent beam for binding pose calculation. To identify amino acids that might be relevant for Yoda binding in the flat state, we quantified the occurrence of all residues located within 3.5 Å of Yoda1 in all detected binding poses (Fig. 2C). This analysis revealed two important things: Firstly, in contrast to Botello-Smith and colleagues who suggested that the Yoda1 binding site is lined by A1718, A2091, and A2094[26], none of the Yoda1 binding poses found here for the curved conformation appeared to involve interactions with A2094 and A1718, and A2091 was only found in 3 from 11 possible docking poses. Closer inspection of the PIEZO1 structure used for modelling (7WLT[16]) and all other previously published PIEZO1 structures (6B3R[12], 6BPZ[13], 5Z10[14], AF-E2JF22-F1-v4[35]), showed that the space between the three afore-mentioned residues is occupied by M1719 in all PIEZO1 cryo-EM structures (Supplementary Fig. 2B), whereas it is accessible in the molecular dynamics simulation-refined structure that was analysed by Botello-Smith et al., where M1719 is flipped upwards and the entire α-helix in which it resides is tilted backwards (Supplementary Fig. 2C). This subtle, yet crucial

structural difference, explains why we did not detect Yoda1 binding poses in the curved conformation that involve the previously described amino acids. Most importantly, however, our analysis highlighted F1715, which appears to participate in Yoda binding in ~80% of all possible docking poses in the flattened conformation, but not in a single binding pose found in the curved state (Fig. 2C), suggesting that it may play a crucial role during channel activation. Notably, F1715 had also been implicated in Yoda1 binding in the flattened state by Jiang and colleagues, who proposed that F1715, together with V1714 and I1696 create a high-affinity Yoda1 binding site using molecular dynamics simulation of an in-silico modelled flat PIEZO1 conformation[27].

Hence, our DoGSite3-based pocket analysis and AutoDock Vina docking pose prediction together with the advanced molecular dynamics simulations of Jiang et al., which also considered the impact of membrane lipids, water molecules and ions on ligand binding, strongly suggest that PIEZO1 presents distinct Yoda1 binding sites in the curved versus flattened conformation.

## Yoda-dependent modulation and activation of PIEZO1 require different binding sites

To directly test the two-binding sites hypothesis, we generated a PIEZO1 double mutant in which F1715 (detected here as well as by Jiang et al.[27]) and its neighbour V1714 (implicated in Yoda binding in the flat state by Jiang et al.) were substituted by alanines (PIEZO1.-V1714A_F1715A) and compared its Yoda sensitivity with that of PIEZO1 wildtype and PIEZO1.A2094W.

Our data as well as the previous work of Botello-Smith et al. indicated some residual sensitivity of the A2094W mutant to high concentrations of Yoda1 (100 μM)[26], but due to the limited solubility of Yoda1, higher concentrations could not be tested and hence it remained unclear if the A2094W mutation solely shifts the $EC_{50}$ to higher concentrations while still allowing maximal activation, which would indicate a reduction in binding affinity, or if the maximal Yoda1-induced response was also altered, which would point at an allosteric effect. We thus utilized the Yoda1 analogue Yoda2, which has improved solubility and potency compared to Yoda1[36], such that concentrations that are several orders of magnitude higher than the $EC_{50}$ on wildtype PIEZO1 can also be tested. Consistent with previous reports, we observed an $EC_{50}$ of 0.03 μM for Yoda2-induced calcium influx assessed by GCamp8 calcium imaging in N2a cells over-expressing wild-type PIEZO1 (Fig. 3A, B). N2a-P1KO cells only expressing GCamp8, did not respond to Yoda2, demonstrating that the observed $Ca^{2+}$ influx was mediated by PIEZO1 (Supplementary Fig. 3A). At high concentrations Yoda2 also activated PIE-ZO1.A2094W ($EC_{50} = 7.03$ μM) and the response magnitudes almost reached the same level as PIEZO1 mediated responses, suggesting that the A2094W mutation indeed alters binding affinity but not Yoda's ability to fully activate PIEZO1. Notably, the maximal response amplitudes and the $EC_{50}$ for activation (0.067 μM) of the putative flat state-specific binding site-2 mutant (PIEZO1.V1714A_F1715A), were indistinguishable from those of wildtype PIEZO1, indicating that binding site-2 is dispensable for the chemical activation of the channel (Fig. 3A, B).

We next considered the Yoda2-dependent modulation of mechanically-evoked currents. To this end, we first examined the effect of increasing Yoda2 concentrations on the pressure-response curve of wildtype PIEZO1, which revealed that as little as 30 nM of Yoda2 is sufficient to significantly shift the pressure sensitivity to less negative pressures (Fig. 3C and Supplementary Fig. 3B). The same concentration also robustly modulated poking-evoked PIEZO1 currents in whole-cell recordings (Fig. 3D). Consistent with our data obtained with Yoda1 (Fig. 1C, D), Yoda2 also markedly modulated mechanically-evoked currents mediated by the activation-deficient PIEZO1.A2094W mutant (Fig. 3C, D). Strikingly, however, the PIE-ZO1.V1714A_F1715A mutant, which exhibits normal Yoda2 sensitivity in calcium imaging experiments (Fig. 3A, B), appeared to be significantly less sensitive to Yoda2-dependent modulation (Fig. 3C, D and Supplementary Fig. 3C–E). Thus, 30 nM Yoda2, which robustly modulated PIEZO1 and PIEZO1.A2094W neither altered the $P_{50}$ values of stretch-evoked currents in pressure-clamp recordings (Fig. 3C) nor the amplitudes and thresholds of poking-evoked currents in whole-cell recordings (Fig. 3D). Higher concentrations of Yoda2 (300 nM, 1 μM and 3 μM) did, however, cause subtle modulation of mechanically-evoked PIEZO1.V1714A_F1715A currents (Supplementary Fig. 3C). Importantly, both mutants were indistinguishable from wildtype PIEZO1 in the absence of Yoda2 with regards to sensitivity (P50 and poking threshold) and single-channel conductance (Supplementary Fig. 3F–H). An attempt to generate a completely Yoda1-insensitive mutant by simultaneously disrupting both binding sites (PIEZO1.V1714A_F1715A_A2094W triple mutant) rendered PIEZO1 completely silent with regards to mechanosensitivity and chemo-sensitivity (Supplementary Fig. 3I, J), thus precluding proper interpretation. In summary, our data shows that mutations in binding site-2 affect the concentration-dependence of Yoda2-induced modulation but not activation, suggesting that this binding site is preferentially involved in the former effect.

## MINFLUX nanoscopy suggests that binding site I but not binding site II is required for Yoda-induced flattening of PIEZO1
The observation that Yoda-mediated activation can be abolished while modulation is preserved—and vice versa—was rather unexpected and contradicted the currently prevailing view about Yoda's mechanism of action, which suggests that activation and modulation both result from Yoda-induced partial flattening of the channel. To shed light onto this conundrum, we sought to directly measure Yoda-induced flattening of the activation and modulation-deficient PIEZO1 mutants in fully intact cells, using 3D-MINFLUX nanoscopy in combination with DNA-PAINT. 3D-MINFLUX/DNA-PAINT achieves spatial resolutions in the nanometer range[37–39], thereby enabling us to resolve the distances between fluorophores attached to the peripheral ends of the blade domains and thus observe PIEZO1 flattening in its native environment (Fig. 4A). To this end, we inserted ALFA-tags[40] into position H86 (extracellular distal end of the blade), which does not alter channel function[29], and imaged cells with 3D-MINFLUX using a DNA-PAINT docking strand-conjugated single domain anti-ALFA nanobody and an Atto655-conjugated DNA-PAINT imager strand (Fig. 4A). This labelling system has a linkage error of only ~2.5 nm, thereby allowing precise position estimates of the distal ends of the blades (Supplementary Fig. 4A). To identify triple-labelled PIEZO1 trimers in MINFLUX localisation data, we computed the Euclidian distance matrix of all localisations and searched for localisation triplets in which the distances between the individual localisations were smaller than 40 nm (i.e., the maximal physically possible distance between two ALFA-tags in the fully flattened state) and that had no other neighbouring signals within 60 nm (see Supplementary Fig. 4B–D for triple-labelled trimer identification rules). Using this approach, we have previously shown that PIEZO1 conformation at rest varies between cellular compartments and is governed by cytoskeletal rigidity[29]. Hence, to rule out confounding effects resulting from variability in cytoskeletal rigidity, we examined the impact of Yoda1 on PIEZO1 flattening in the presence of cyto-chalasin-D, which disrupts the cytoskeleton and drives most channels into the curved state (Supplementary Fig. 4E). In agreement with our previous study[29], 50 μM Yoda1, which robustly activates PIEZO1 in calcium imaging experiments (Fig. 4C), induced significant channel flattening, increasing the mean interblade distance by 2.19 nm from $21.69 \pm 5.52$ nm (CTL) to $23.88 \pm 5.5$ nm (Yoda1; Fig. 4D). The activation-deficient PIEZO1.A2094W mutant (Figs. 1B and 4C), by contrast, showed no sign of Yoda1-induced flattening with interblade distances of $20.86 \pm 5.11$ nm under control conditions and $20.86 \pm 6.07$ nm in the presence of Yoda1 (Fig. 4E), whereas the interblade distance of PIEZO1.V1714A_F1715A increased from $21.48 \pm 5.65$ nm to $24.49 \pm 5.49$ nm (Fig. 4F).

It is important to emphasize that the relatively large variability of the interblade distance distributions does not reflect a lack of precision of MINFLUX nanoscopy or a large linkage error of the DNA-PAINT labelling system. Instead, as we and others have previously shown using 3D-MINFLUX/DNA-PAINT and single-particle cryo−light microscopy, in combination with labelling techniques with different linkage errors, the broad distribution of interblade distances shows that PIEZO1 adopts the full spectrum of conformational states with different interblade distances in its native environment, ranging from the curved conformation with three handshakes between the peripheral blades (smallest possible interblade distance) to the fully flattened conformation (largest possible interblade distance)[28,29,41,42]. Accordingly, Yoda1 does not equally shift all interblade distances to larger values but solely changes the distribution of interblade distances within the physically possible margins (curved with three handshakes−fully flattened).

Together, these data show that Yoda1's ability to flatten PIEZO1 indeed correlates with its ability to activate the channel (Fig. 4C). Interestingly, however, our data also suggest that Yoda1-induced PIEZO1 flattening is neither required (i.e., PIEZO1.A2094W is modulated but not flattened) nor sufficient (i.e., PIEZO1.V1714A_F1715A is flattened but not modulated) for the modulation of mechanically-evoked currents.

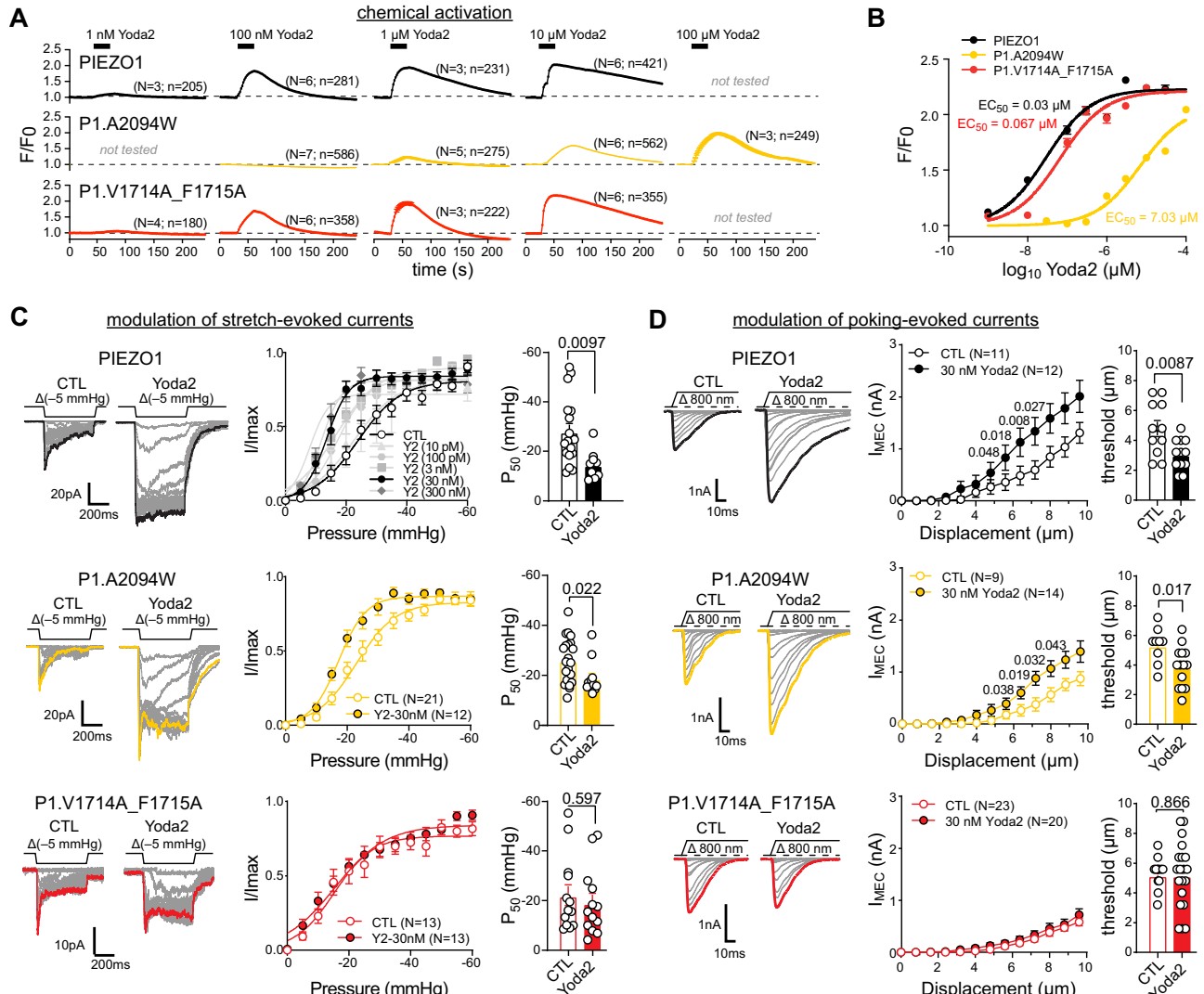

**Fig. 3 | Yoda-dependent activation and modulation involve different binding sites. A** Time courses of mean normalized fluorescence ± SEM (F/F$_0$) of Yoda2-evoked calcium influxes in N2a-P1KO cells co-transfected with jRGECO1a and PIEZO1 (black), P1.A2094W (yellow) and P1.V1714A_F1715A (red), respectively. "*N*" denotes the numbers of coverslips and "*n*" denotes the total number of cells. **B** Yoda2 dose-response curves, i.e., max. F/F$_0$ (mean ± SEM) vs. Yoda2 concentration, and sigmoidal fits with corresponding EC$_{50}$ values. *N*-numbers for (**B**) are the same as in (**A**). N-numbers not provided in (**A**) are: N/n, PIEZO1 (10 nM, 5/321; 300 nM, 3/257; 3 μM, 3/241; 30 μM, 3/238), P1.A2094W (30 nM, 3/277; 300 nM, 3/190; 3 μM, 3/179; 30 μM, 3/250), P1.V1714A_F1715A (10 nM, 4/281; 300 nM, 3/224, 3 μM, 3/267; 30 μM, 3/237). **C** The left panel shows representative traces of stretch-evoked PIEZO1 (top), A2094W (middle) and V1714A_F1715A (bottom) currents in the absence (CTL) and presence of 30 nM Yoda2. Middle panel shows comparisons of normalized pressure-response curves in the absence (CTL) and presence (Y2) of 30 nM Yoda2, and additional concentrations for PIEZO1 (CTL, *N* = 17; 10 pM, *N* = 5,

100 pM, *N* = 9; 3 nM, *N* = 7; 30 nM, *N* = 9; 300 nM, *N* = 12). Symbols represent means ± SEM normalized peak current amplitudes (I/I$_{max}$) and N-numbers indicated in the graphs refer to the number of different cells from which currents were recorded. Pressure-response relationships were fitted separately for each cell with a Boltzmann equation and the mean ± SEM pressures at which half maximal activation was reached (P$_{50}$) are compared in the bar graphs in the right panel using two-sided Student's *T*-test for PIEZO1 and two-sided Mann–Whitney test for A2094W and V1714A_F1715A. **D** Representative traces of poking-evoked whole-cell currents from PIEZO1 (top left), A2094W (middle left) and V1714A_F1715A (bottom left) without (CTL) and with 30 nM Yoda2. Displacement-response curves of mean ± SEM peak current amplitudes (middle) and comparison of the mean ± SEM threshold with two-sided unpaired *t*-test (PIEZO1, A2094W) and Mann–Witney test (V1714A_F1715A) (right). Response amplitudes at different displacements were compared using two-sided Mann–Whitney test without adjustments for multiple comparison and *P*-values are provided next to symbols in graphs.

## Discussion

The currently prevailing mechanistic framework regarding Yoda's mode of action proposes that Yoda1 acts as a molecular wedge that binds to a hydrophobic pocket at the THU8−9 interface in the curved conformation, thereby driving PIEZO1 into a partially flattened state that facilitates mechanical activation, such that even basal membrane tension is sufficient to activate the channel[25,26]. Recently, this model was refined using in-silico molecular dynamics simulations, which suggested that Yoda1 might transition to another binding site with higher affinity in the PIEZO1 open conformation[27].

Here we made several important observations that corroborate some aspects of this mechanistic model, while they challenge and refine others: Firstly, using patch-clamp recordings and GCamp8 calcium imaging, we showed that mutating an essential residue in the previously proposed Yoda binding site (A2094W)[26] abolishes Yoda-induced activation but preserves modulation of mechanically-evoked PIEZO1 currents (Fig. 1), whereas substitution of V1714 and F1715 with alanine, has the diametrically opposing effect−i.e., it preserves Yoda-induced calcium influx but abolishes modulation (Fig. 3). Secondly, by measuring the interblade distances (i.e., a surrogate measure for

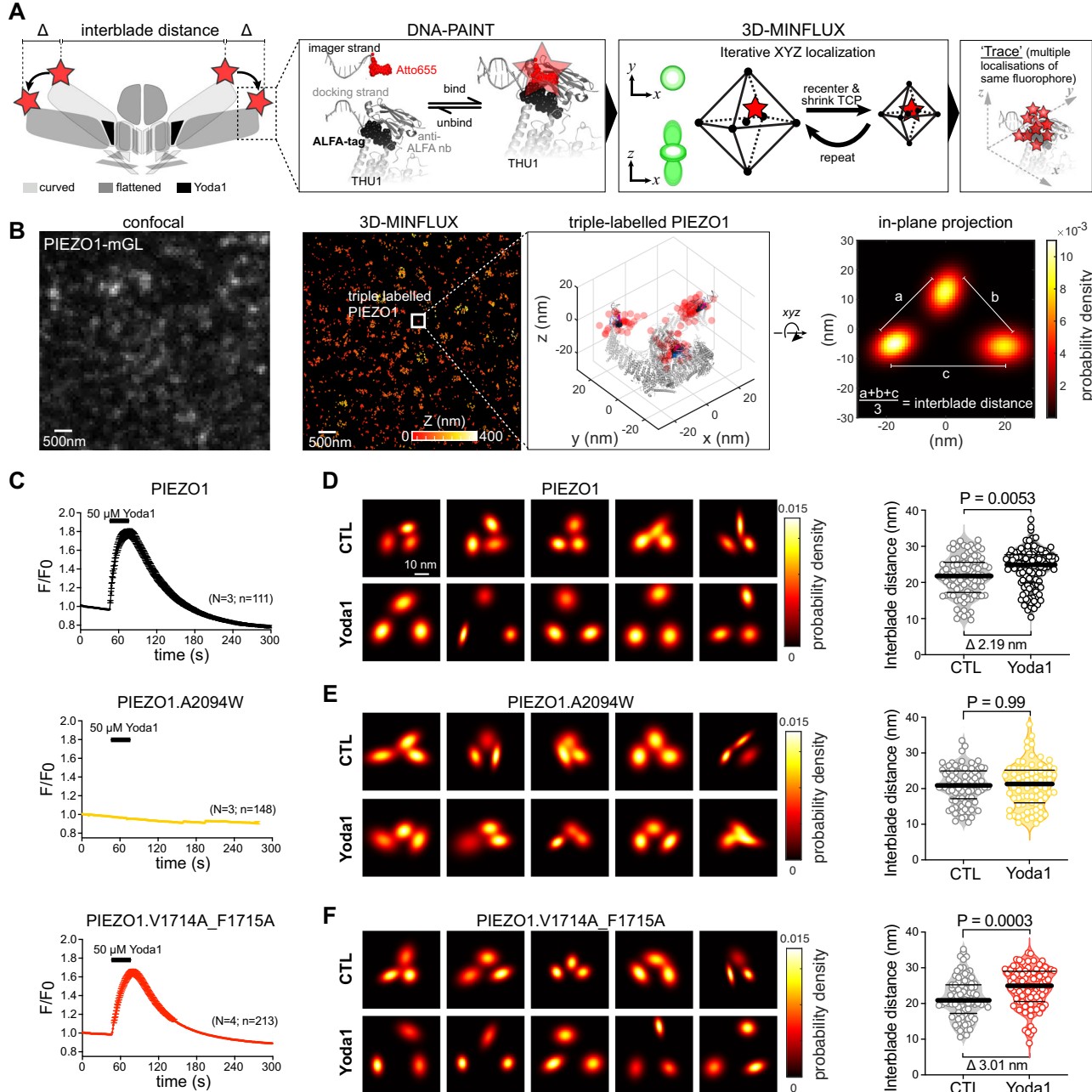

**Fig. 4 | Direct observation of Yoda1-induced conformational changes in wild-type as well as activation- and modulation-deficient PIEZO1 mutants with 3D-MINFLUX. A** Cartoon depicting the overall strategy to resolve Yoda1-induced conformational changes (flattening) measured through changes in interblade distance. Insets illustrate the labelling of PIEZO1 with ALFA tag inserted after H86, and the DNA-PAINT method (left). Individual bound fluorophore is located with high precision in 3 dimensions via 3D-MINFLUX and its iterative process (middle), leading to multiple localisations of the same molecule. **B** Workflow example showing confocal image of a PIEZO1-ALFA-mGL expressing N2a-P1KO cell (left) and corresponding 3D-MINFLUX localizations (right), colored by Z range. Inset shows a triple-labelled PIEZO1, with the 3D scatter plots of the raw localizations and a superimposed cryo-EM structure. The 2D in-plane projections of the 3D data were fitted with a bivariate Gaussian distribution, with their probability densities,

enabling determination of the average interblade distance (right). **C** Time courses of mean normalized fluorescence ± SEM (F/F₀) of 50 μM Yoda1-evoked calcium influxes in N2a-P1KO cells co-transfected with jRGECO1a and PIEZO1 (black), P1.A2094W (yellow) and P1.V1714A_F1715A (red), respectively. "*N*" denotes the numbers of coverslips in which the effect of Yoda1 was tested and "*n*" denotes the total number of cells that were tested. **D–F** In-plane projections of representative trimers examples of PIEZO1 (**D**), A2094W (**E**) and V1714A_1715A (**F**) from cells treated with cytochalasin-D (CTL) or with cytochalasin-D and Yoda1 (50 μM) (left). Comparison of the mean ± SEM interblade distance of the identified trimers after addition of Yoda1 for PIEZO1 (**D**, *N* = 93 and 110), A2094W (**E**, *N* = 86 and 82) and V1714A_1715A (**F**, *N* = 91 and 96), with two-sided Student's unpaired *t*-test (right). *N*-numbers in (**D–F**) represent the numbers of biological replicates (i.e., numbers of triple-labelled PIEZO channels that were detected).

channel flattening) in PIEZO1 using 3D-MINFLUX nanoscopy, we demonstrate that Yoda fails to flatten the activation-deficient A2094W mutant, whereas it robustly flattens the modulation-deficient PIEZO1.V1714A_F1715A mutant (Fig. 4). Together, these data demonstrate that Yoda's ability to flatten PIEZO1 in the absence of mechanical

stimuli is neither required (i.e., A2094W is modulated but not flattened) nor sufficient (i.e., V1714A_F1715A is flattened but not modulated) to modulate mechanically-evoked currents, thereby challenging the previously proposed mode of action of Yoda1 and its analogues with regards to modulation. With regards to Yoda-mediated activation,

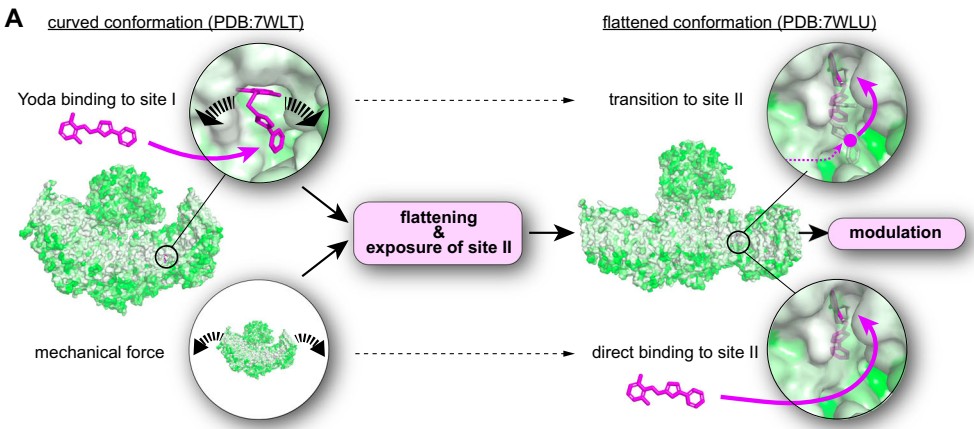

**Fig. 5 | Mechanistic model of Yoda's mode of action. A** In the closed state PIEZO1 adopts a curved conformation (PDB:7WLT) in which binding site-I is accessible. Yoda binding to binding site-I induces a conformational change that is characterized by downward tilting of the blade domains, resulting in a flattened conformation (top left and Fig. 4D, E). Flattening also occurs in the presence of mechanical force (bottom left and refs. 16,17). In the flat conformation binding site-II becomes accessible, such that Yoda can either transition from site-I to site-II (top right) or directly binds to site-II (bottom right).

our data is, however, consistent with the current model as it shows a correlation between activation and flattening—i.e., wildtype PIEZO1 and PIEZO1.V1714A_F1715A are activated and flattened, whereas PIEZO1.A2094W is neither activated nor flattened. So how can modulation work without flattening if the supposed binding site-2 only becomes available in the flat state? The answer is it does not. It is important to note that PIEZO1 also flattens in response to mechanical stimulation—as suggested by computational modelling[12,43] and shown by high-speed atomic force microscopy[17]—such that binding site-2 becomes available, which explains why Yoda1 can still modulate PIEZO1.A2094W despite lacking the primary activation-specific Yoda binding site. In summary, our data support a two-sites induced-fit-like mechanism where Yoda1 first engages binding site-1 to promote blade flattening and pore opening and then transitions to a deeper binding site to modulate the mechanical activation threshold and inactivation kinetics (Fig. 5).

While our data together with the work of Jiang et al.[27] conceptually support a two-site induced-fit-like mode of action of Yoda1, they do not allow definitive conclusions about the exact role of A2094 and F1715. Thus, it remains unclear whether these residues actively participate in Yoda binding in the curved and flat state, respectively, or if they are solely involved in allosterically coupling conformation changes induced by Yoda binding elsewhere. Previous studies have shown that point mutations can differentially affect Yoda-, stretch- and poking sensitivity in PIEZO1 and PIEZO2, suggesting that different types of stimuli engage different intramolecular force transmission pathways to activate PIEZOs, thereby indicating that one should be cautious when interpreting site-directed mutagenesis data with regards to Yoda binding[14,18,20,26,44–46].

Regarding the precise role of A2094, we obtained conflicting results. On the one hand, we found that substitution of A2094 with tryptophane shifts the $EC_{50}$ for Yoda2 to higher concentrations but does not alter the maximal response amplitudes in calcium imaging experiments (Fig. 3A), which indicates that this mutation only alters the Yoda binding affinity but not the allosteric coupling mechanisms required for Yoda-induced channel activation and therefore suggests an active role of A2094 in Yoda binding, as previously proposed by Botello-Smith and colleagues[26]. Our docking pose analysis, on the other hand, argues against a direct participation in Yoda binding as we did not detect A2094 within 3.5 Å of any predicted docking pose in the curved state (Fig. 2C). It should, however, be noted that the docking tools used here do not account for the membrane environment and rely solely on static protein conformations, which could result in false-

negative as well as false-positive binding sites and binding poses. Moreover, detailed inspection of all five previously published curved PIEZO1 structures revealed that the space between A2091, A2094, and A1718, which supposedly line the curved state binding site, is occupied by M1719 and thus not accessible for Yoda (Supplementary Fig. 2A). A2094 is located in close proximity to the binding pocket detected here (Fig. 2B) and thus it is possible that the tryptophane substitution nevertheless selectively—yet indirectly—alters Yoda binding affinity without affecting allosteric coupling, for example by limiting access to the binding site. Interestingly, in the structure analysed by Botello-Smith et al.[26], which derived from the cryo-EM structure published by Guo & MacKinnon (PDB:6B3R)[12] and was refined considering the presence of membrane lipids, water molecules and ions, however, M1719 and the α-helix in which it resides, assume a completely different orientation such that Yoda can interact with A2094 (Supplementary Fig. 2B). It is pointless to speculate about which structural model is more accurate because definitive clarity could only be obtained by resolving the Yoda-bound curved PIEZO1 structure, which was far beyond the scope of this study. With regards to the role of F1715 evidence for an activate participation in Yoda binding in the flat conformation is more compelling. Thus, F1715 was not only detected as a putative interaction partner of Yoda1 in almost all binding poses predicted here (Fig. 2C) but was also implicated in Yoda1 binding by Jiang and colleagues who used a completely different approach[27]. Moreover, the F1715A mutation selectively shifted the concentration-dependence of Yoda-dependent modulation to higher concentrations (Fig. 3C, D and Supplementary Fig. 3C) but did not alter mechanosensitivity (CTL in Fig. 3C, D) and chemosensitivity (Fig. 3A) per se nor did it alter PIEZO1's ability to flatten (Fig. 4E), suggesting a specific role of F1715 in Yoda binding in the flat state. Regardless of remaining uncertainties regarding the exact roles of A2094 and F1715 and the lack of information about the role other amino acids in the proposed binding sites, our data nevertheless strongly support a refined mechanistic model for Yoda's mode of action in which activation and modulation are mediated by distinct binding sites.

In recent years, there have been intensive efforts to develop PIEZO1 specific agonists[24], which have led to the discovery of several drugs such as Yoda1[23], Jedi1 and Jedi2[18], Yoda2[36], Yaddle1[47], and CMPD15[27]. A common feature of these drugs is that they both activate and modulate PIEZO1, which complicates the mechanistic interpretation of their effects and thus limits their use for research and clinical purposes, because it is impossible to distinguish between effects

resulting from Yoda1-induced calcium influx, which merely indicates the presence of PIEZO1 in a specific cell type, and effects caused by modulation of the channel, which would indicate a true role of PIEZO1-dependent mechanotransduction in a specific physiological process. By demonstrating that PIEZO1 activation and mechanical sensitization by Yoda compounds are separable processes that depend on two distinct ligand-binding sites, our study provides a framework that can guide the design of novel drugs that selectively target one site without engaging the other, which may eventually lead to the development of agonists and modulators with tailored functional profiles for research and therapeutic applications.

## Methods

### Generation of PIEZO mutants
Previously validated plasmids expressing mouse PIEZO1-mScarlet (C-ter insertion of mScarlet with a SG linker) and PIEZO1-ALFA-mGL (insertion of ALFA tag after H86 and mGreenLantern at C-ter with a SG linker), which do not affect PIEZO1 function[29], were used as the initial template to generate the mutants of the present work, using a similar strategy described elsewhere[45,46]. For calcium imaging experiments, the plasmids expressing jGCaMP8m and jRGECO1a were purchased from Addgene (#162372 and #100854). Point mutations were introduced by PCR (primers from Sigma, see Supplementary Table 1 for primers sequence, template and obtained constructs) using KAPA HiFi polymerase (Roche). PCR reactions were digested with DpnI (New England Biolabs, 37 °C, 1 h) and column purified with standard kits (NucleoSpin from Macherey-Nagel or PureLink from Invitrogen) before being transformed in electrocompetent Dh5a bacteria (Invitrogen) and grown at 37 °C overnight. Selected clones were entirely sequenced (Eurofins) to ensure that no unwanted mutation was present.

### Cell culture
Mouse neuroblastoma Neuro-2a PIEZO1-Knockout cell line (N2A-P1KO) was generated previously from Neuro-2a ATCC CCL-131 (a gift from G.R Lewin[30]). Cells were grown at 37 °C with 5% $CO_2$ in a 1:1 mixture of Dulbecco's Modified Eagle Medium and Optimal Minimal Essential Medium, with 10% Fetal Bovine Serum, 2 mM L-glutamine and 1% peniciline/streptomycine (all from ThermoFisher). Cells were seeded on methanol- and acid-washed glass coverslips and coated with poly-L-lysine (PLL, Sigma). 12 mm diameter coverslips were used for patch-clamp recordings and calcium imaging, and #1.5 and 18 mm diameter for Minflux imaging. Cells were transfected one or two days after plating using polyethylenimine (PEI, Linear PEI 25 K, Polysciences). For one 12 mm coverslip, 7 μl of a 360 μg/ml PEI solution is mixed with 9 μl PBS. Plasmid DNA is diluted in 20 μl PBS (0.6 μg/coverslip) and then the 16 μl PEI-PBS solution is added to the DNA solution. After at least 10 min of incubation at room temperature, the DNA-PEI mix is added drop by drop and mixed by gentle swirling. For calcium imaging experiments, 0.3 μg/coverslip of jGCaMP8m or jRGECO1a plasmid is also added. For a 18 mm coverslip (Minflux), 2.0 μg DNA is used and PBS/PEI volumes are adjusted accordingly. Twenty-four hours after transfection, medium is replaced by fresh one and cells are then used within 24 h to 48 h.

### Electrophysiology
PIEZO mutants were characterized in patch-clamp assays and compared to control PIEZO1-mScarlet or PIEZO1-ALFA-mGL. Mechanically activated currents were recorded at room temperature using EPC10 amplifier with Patchmaster software (HEKA Elektronik). Borosilicate patch pipettes (2–6 MΩ for whole-cell, 1.5–3.5 MΩ after fire-polishing for cell-attach) were pulled with a P-97 Flaming-Brown puller (Sutter Instrument). For whole-cell patch-clamp, intracellular buffer contained the following: 125 mM κ-gluconate, 7 mM KCl, 1 mM $MgCl_2$, 1 mM $CaCl_2$, 4 mM EGTA, 10 mM HEPES (pH 7.3 with KOH). For cell-attached

recordings: 130 mM NaCl, 5 mM KCl, 1 mM $MgCl_2$, 1 mM $CaCl_2$, 10 mM HEPES, 10 mM TEA-Cl (pH7.3 with NaOH). The control bath solution for whole-cell contained the following: 140 mM NaCl, 4 mM KCl, 1 mM $MgCl_2$, 2 mM $CaCl_2$, 4 mM glucose and 10 mM HEPES (pH 7.4 with NaOH). For cell-attach recordings, the bath solution contained: 140 mM KCl, 1 mM $MgCl_2$, 2 $CaCl_2$, 10 Glucose, 10 HEPES (pH7.4 with KOH). Cells were held at a holding potential of −60 mV (whole-cell and cell-attach). To determine conductance of a given mutant, holding potential was changed from −140 to −40 mV (Δ20 mV). To study the modulatory effect of Yoda1 and Yoda2 on mechanically activated PIEZO1 currents, drugs were diluted and added into the pipette solution at the indicated concentration.

Mechanical stimulation in whole-cell experiments was done with a series of 15 mechanical stimuli in 0.8 μm increments with a fire-polished glass pipette (tip diameter 2–3 μm) that was positioned opposite to the recording pipette, at an angle of approximately 45° to the surface of the dish and moved with a velocity of 1 μm/ms by a piezo driven micromanipulator (Preloaded Piezo actuator P-840.20, Physik Instrumente). Negative pressure stimuli in cell-attach experiments were applied for 500 ms with the High-Speed Pressure Clamp device (HSPC; ALA Scientific Instruments), with −5 mmHg increments up to −80 mmHg. A pre-pulse of +5 mmHg was applied before negative-pressure stimuli to improve recovery from inactivation[48]. For I/V and single-channel conductance experiments, pressure stimulus was adjusted on a cell-by-cell basis to optimally evoke single-channel openings.

The evoked whole cell currents were recorded with a sampling frequency of 200 kHz and filtered with 2.9 kHz low-pass filter. Pipette and membrane capacitance were compensated using the auto function of Patchmaster. Recordings with excessive leak currents, unstable access resistance and cells which giga-seals did not withstand at least 7 consecutive mechanical steps stimulation were excluded from analyses. Mechanical thresholds of PIEZO currents were determined by measuring the mechanical stimulus that evoked the first sizeable peak current, defined as the point in which the current significantly differed from the baseline (more than 6 times the standard deviation of the baseline). The inactivation time constants ($\tau_{inact}$) were measured by fitting the mechanically activated currents with a single exponential function ($C1 + C2*e^{(-(t-t0)/\tau_{inact})}$, where C1 and C2 are constants, $t$ is time and $\tau_{inact}$ is the inactivation time constant. For each cell, only peak currents between 100 and 1500 pA were used for $\tau_{inact}$ calculation and averaged from cell to cell.

The evoked cell-attach currents were recorded with a sampling frequency of 50 kHz and filter with a 2.9 kHz low-pass filter. Maximal pressure-evoked currents over the course of a given stimulus (I) were normalized to the absolute maximal response of the cell at any pressure ($I_{max}$). Normalized pressure-response curve ($I/I_{max}$) from individual cells were fitted with a Boltzmann sigmoid to determine individual P50 (in mmHg).

Single-channel amplitudes at a given holding potential (−140 mV to −40 mV, 20 mV steps) were determined as the difference between the peaks of the Gaussian fits of the trace histogram over multiple 1 s segments. Unitary conductance was determined from the linear regression fits of the I/V plot of individual cells. Recordings with excessive leak currents or unstable baseline were excluded. Recordings that displayed non-inactivating responses or unstable openings were also not used for further I/V analyses. All electrophysiology analysis was performed in IgorPro (Wavemetrics) using custom scripts.

### Calcium imaging
N2A-P1KO cells were cultured as described above and transfected with PIEZO1-mScarlet or PIEZO1-ALFA-mGL or associated PIEZO1 mutants (0.6 μg per coverslip) together with jGCaMP8m (if using a PIEZO-mScarlet) or jRGECO1a (if using a PIEZO-ALFA-mGL) (0.3 μg per coverslip). Cells were washed once with PBS and incubated with a calcium

imaging buffer containing 140 mM NaCl, 4 mM KCl, 1 mM $MgCl_2$, 3 mM $CaCl_2$, 4 mM glucose and 10 mM HEPES (pH 7.4 with NaOH). Fluorescent images were acquired every two seconds (500 ms exposure time) on an Olympus BX40 upright microscope equipped with standard Quad filter (Chroma), fluorescent lamp (HBO 100) and shutter (Lambda 10-2, Sutter Instrument) with a 20× water-immersion objective (UMPLFLN 20XW, Olympus), visualized with a CoolSnap HQ2 camera (Photometrics) and acquired with the MetaFluor software (Molecular Devices). Perfusion and fast solution exchanged was achieved with a gravity-driven perfusion system (ValveLink8.2, AutoMate Scientific). Cells were first perfused with control solution for at least 30 s before being exposed to Yoda1 (Sigma, 10, 50 or 100 μM, diluted in calcium imaging buffer) or Yoda2 (Tocris, from 1 nM to 300 μM) for 30 s, followed with a washout with control solution. Only one field of view per coverslip was used to avoid Yoda leakage. Only cells that are double positive for PIEZO1 and the calcium indicator were considered for analysis. Cells were segmented in ImageJ and the time course of normalized fluorescence ratio ($F/F_0$) was calculated as the ratio between the jGCaMP8m/jRGECO1a fluorescence intensity (a.u) at a given time (F) and the average fluorescence intensity per PIEZO-transfected cell averaged over a 10 s interval during the initial control perfusion ($F_0$). Within the Yoda1 perfusion time, the maximal $F/F_0$ ratio per cell was extracted, the average for each concentration calculated and fitted with sigmoid curve for $EC_{50}$ determination.

### Preparation of samples for 3D-MINFLUX/DNA-PAINT imaging
N2A-P1KO cells were plated on 18 mm coverslips as described above and transfected with PIEZO1-ALFA-mGL and associated mutants. Two to three days after transfection, cells were processed for treatment. For cytochalasin D (cytoD, Sigma) experiments, cells were incubated for 20 min at 37 °C with 2 μM cytoD diluted in culture medium, which was kept after in the fixative solution. For Yoda1 experiments, based on previous experiments[29], cells were first incubated with cytoD alone for 17 min and then 50 μM Yoda1 was added for the last 3 min at 37 °C, which was kept after in the fixative solution. Cells were then fixed in a mixture of paraformaldehyde (1%) and glutaraldehyde (0.05%) that contained also the appropriate treatment (no drug control, cytoD or cytoD + Yoda1). Fixatives were quenched for 10 min with 5 mM $NaBH_4$ and then with a mix of 50 mM glycine and 50 mM $NH_4Cl$, all diluted in PBS. After additional washing with PBS, samples were blocked with Antibody Incubation buffer (Massive Photonics) for 30 min at room temperature. The cells were incubated for 1 h at room temperature or overnight at 4 °C with Massive-Tag-Q Anti-ALFA nanobodies conjugated with a DNA docking strand (both from Massive Photonics) diluted at 1:100 (Anti-ALFA) in Antibody Incubation buffer. Cells were washed thrice with 1× Washing buffer (Massive Photonics) and then post-fixed for 5 min. Fixatives were quenched and washed as described above. Samples were then incubated for 10 min with 200 μl of gold nanoparticles for future stabilization (gold colloid 250 nm, BBI Solutions). Unbound nanoparticles were removed with several PBS washes and remaining nanoparticles were further stabilized by incubating the samples with PLL for at least 1 h at room temperature. Cells were then washed thrice with PBS before mounting. Labelled samples were used within 2 days. Corresponding DNA-PAINT imagers (Massive Photonics) conjugated to Atto 655 were freshly diluted in Imaging buffer (Massive Photonics) for a final concentration of 1 - 2 nM (Atto 655, Imager sequence #3). A drop of imager dilution was added into a cavity slide and coverslips were mounted and sealed with Picodent Twinsil (Picodent).

### DNA-PAINT 3D Minflux imaging
Minflux imaging was performed on an Abberior MINFLUX commercial microscope built on an inverted IX83 microscope with a 100× UPlanXApo objective (Olympus) and using Imspector Software (Abberior Instruments). Alignment and calibration of the excitation beam

pattern and position of the pinhole was performed daily using fluorescent nanoparticles (Abberior Nanoparticles, Gold 150 nm, 2 C Fluor 120 nm, Abberior Instruments). PIEZO-ALFA-mGL transfected cells were identified with a 488 nm confocal scan and focus was set on the coverslip-cell surface interface. The transient binding of imagers with Atto 655 was quickly verified with fast 640 nm confocal scan. At least 2 gold fiducials were present in the field of view and used by the active-feedback stabilization system of the microscope (IR 975 nm laser, Cobolt, and CCD camera, The Imaging Source), resulting in a precision below 1 nm in all three axes and being stabled for hours. A ROI of approximately $3 \times 3$ to $5 \times 5$ μm (and up to $8 \times 8$ μm for some overnight recordings) was selected. Laser power in the first iteration was set at 16% laser power and pinhole was set to 0.83 A.U. Final laser power in the last iteration is scaled up by a factor of six. ROIs were imaged for at least 2 h and up to overnight (~12 h) using the standard 3D Minflux sequence (Supplementary Table 2). Detection for Atto 655 signals was performed with two avalanche photodiodes channels (650–685 nm and 685–720 nm) that were pooled.

### Minflux data analysis
Minflux analysis was performed as described before[29]. Valid localizations from Minflux final iterations were exported from Imspector software as MATLAB files. Custom Matlab scripts were then used for post-processing and filtering of the data (https://github.com/StefanLechnerUKE/PIEZO1_Yoda1_MINFLUX), as well as subsequent operations and data visualization. Data filtering involved first a cfr filter (center frequency ratio, directly implemented during the measurement, set at 0.8) and an efo filter (effective frequency at offset, kHz, retrieved for each individual valid locations) to filter-out potential multiple emitters. Then, localizations from the same emission trace, i.e., with the same trace identification number (TID), having a standard deviation of more than 10 nm in the x, y, z axes and less than 3 localizations were excluded. Filtered traces were trimmed of their first 2 localizations, as they are often apart from the rest and the majority of the localization cloud. Remaining traces were further corrected for the refractive index mismatch between the coverslip and the sample, applying a scaling factor of 0.7 for all traces in the z dimension[38]. The center of mass for each individual filtered trace was then calculated. Due to the nature of DNA-PAINT, different traces originating from repeated detection of the same PIEZO protomer were identified using DBSCAN clustering with minPoints of 2 and epsilon of 8 nm. This cutoff was selected based on the localisation precision of MINFLUX and the possible ALFA-tag flexibility (Fig. 4A, B). The position of the protomers that were detected multiple times was determined by calculating the mean coordinate of the localisations clustered by the DBSCAN algorithm. PIEZO1 trimers in which all three protomers were labelled and detected by MINFLUX were identified by searching for three adjacent traces that were less than 40 nm apart, which we assumed is the maximum distance two Atto-655 molecules bound to the same PIEZO1 trimer can possibly have based on available flattened PIEZO1 structures, and that had no other neighbouring traces within a distance of 60 nm (Supplementary Fig. 4C, D). Moreover, only trimers in which the maximum interblade angle was smaller than 120° were considered. Interblade distance for a given trimer was calculated as the average of the three protomers distance. Further visualization of trimers was done by a 2D in-plane projection of the raw localizations for each protomer, followed by a fit with a bivariate Gaussian distribution and displayed with its probability density.

### Structure modelling, docking and data visualization
Pockets detection was performed with DoGSite3 available on the ProteinPlus server[32]. A reduced segment of the curved (PDB:7WLT) and flat (PDP: 7WLU) PIEZO1 was used, containing residues 1299 to 2104 from one chain, together with Yoda1. Modelling of Yoda poses was done with SwissDock using AutoDock Vina[33,34], with the shortened

curved and flat PIEZO1 structures described above. In some cases, and only for visualization purposes, a full length PIEZO1 was generated. The predicted AlphaFold structure of mouse PIEZO1 (AF-E2JF22-F1-v4) was aligned and superposed onto the experimentally determined curved (PDB 6B3R) PIEZO1 structure to visualize the unresolved peripheral blade. The final constructs display the experimentally resolved structure with the added missing blade parts from AlphaFold as transparent color. A full length PIEZO1 trimer bearing an ALFA tag at position H86 with its nanobody (PDB 6I2G), DNA docking site, imager and fluorophore was also generated. All subsequent modification operations and visualization were performed in PyMol (Schrodinger). Plasmid design and visualization was performed with SnapGene (version 7, Dotmatics). All the other data, graphics and schematics were elaborated and visualized in Matlab, IgorPro, Illustrator (Adobe) and GraphPad Prism (version 10, GraphPad Software).

### Statistical tests and reproducibility

All experiments in this study were performed independently at least three times, yielding similar results. For Minflux experiments, data are from at least 4 cells from at least 3 independent experiments. No statistical method was used to predetermine sample size. Experiments were not randomized and investigators were not blinded during experiments and analysis. Data distribution was systematically evaluated using D'Agostino–Pearson test and parametric or nonparametric tests were chosen accordingly. The statistical tests that were used, the exact P-values and information about the number of replicates are provided in the figure or the corresponding legends.

### Reporting summary

Further information on research design is available in the Nature Portfolio Reporting Summary linked to this article.

## Data availability

The MINFLUX raw data generated in this study and corresponding confocal images have been deposited together with the custom written Matlab analysis code on Github [https://github.com/StefanLechnerUKE/PIEZO1_Yoda1_MINFLUX] and Zenodo [https://doi.org/10.5281/zenodo.17700348][49]. The patch-clamp and calcium imaging data generated in this study are provided in the Source Data File. Previously published PDB files that were used for Yoda docking pose analysis are available at: 7WLT, 7WLU, 6B3R, 5Z10, 6BPZ, Source data are provided with this paper.

## Code availability

The custom Matlab codes for Minflux analysis are available at [https://github.com/StefanLechnerUKE/PIEZO1_Yoda1_MINFLUX] and at Zenodo https://doi.org/10.5281/zenodo.17700348[49].

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

## Acknowledgements
We thank Dr. Antonio Failla and the team of the imaging core facility at UKE Hamburg (DFG Research Infrastructure Portal #RI_00489) for technical assistance with the MINFLUX microscope. Funding for the MINFLUX microscope was awarded by the Hamburgische Investitions-und Förderbank (IFB, grant no. 51164232) under the Operational Programme Hamburg ERDF 2014–2020, REACT-EU of the European Regional Development Fund (ERDF). We also thank Ms. Kirsten Pfeiffer-Drenkhahn and Mr. Haider Al-Marsoomi for assistance with calcium imaging, cloning and cell culture. This work was funded by the DFG grant LE3210/3-3 awarded to S.G.L.

## Author contributions
C.V. performed all Minflux experiments as well as patch-clamp recordings, calcium imaging and molecular biology. L.R. and N.Z. performed additional patch-clamp recordings and calcium imaging experiments. C.V. and S.G.L. wrote analysis scripts and performed data analysis. S.G.L. acquired funding, wrote the manuscript, visualized data and supervised the project.

## Funding

## Competing interests
The authors declare no competing interests.
