## [Transparent Peer Review file · Nature Communications]

3D-MINFLUX nanoscopy reveals distinct allosteric mechanisms for activation and modulation of PIEZO1 by Yoda1

Corresponding Author: Professor Stefan Lechner

Version 0:

Reviewer comments:

Reviewer #1

(Remarks to the Author)

The authors present robust mutational evidence that two well-known effects of Yoda1 on Piezo1, i.e. spontaneous calcium influx in resting cells and a modulation of mechanically-activated currents, are mediated by separate molecular allosteric mechanisms, likely from the ability of Yoda1 to bind at two distinct sites located within the same inter-repeat binding region of the mechanosensory domain. MINFLUX experiments further reveal that the ability of Yoda1 to spontaneously open the channel correlates with its ability to increase the degree of flattening of this domain, a conformational change known to promote channel opening. The author's molecular interpretation makes sense: Yoda1 binds in the upper site when the channel is mostly closed in resting cells, inducing a partial flattening of the channel and increasing its open probability. The presence of a mechanical stimulus may prompt Piezo to flatten to a higher degree, allowing Yoda to occupy a deeper site and modulate biophysical properties of Piezo currents through a yet unknown mechanism.

Major points:

The authors claim that the presence of a secondary, deeper Yoda binding site proposed by Jiang et al. in 2023 has never been tested experimentally (lines 74,136), was exclusively based on in silico data (line 136) and remains speculative (line 287). This is not true. Jiang et al. virtually docked a virtual compound library onto this deeper binding site in their PIEZO1 open state model, which led them to discover two novel activators after experimentally screening ~150 candidates (given the fact that Yoda1 and Jedi activators were randomly discovered from blindly screening millions of molecules, the odds of randomly identifying two unrelated chemical activators from ~150 compounds are close to zero). One might argue that this experimental evidence is indirect (which it is), but the mutagenesis approach used by the authors and by Botello-Smith et al. to probe the Yoda binding site(s) is also indirect, as mutations can have allosteric effects on a remote binding site (as pointed out by the authors lines 319-321). Hence, the two-sites model, first proposed and experimentally tested by Jiang et al., represents the most up-to-date tested model for Yoda activation. The authors ought to introduce this model as early as in the abstract and rearrange their narrative throughout their manuscript in order to properly contextualize their work with the current literature. For example, the subtitle "Computational modelling reveals an alternative Yoda1 binding site that becomes accessible in the flattened state" suggests that the authors have identified a new Yoda1 binding site, while this should be viewed as an independent validation of the Jiang et al. study.

The authors show that the inability of the 2094W mutant to elicit a calcium influx in the presence of Yoda is likely caused by a loss of binding affinity, as evidenced by an increase of EC50 by two orders of magnitude with no dramatic change in the maximal amplitude of fluorescence signals (Fig 3B). I wonder whether a similar loss of affinity also explains the lack of Yoda modulation in the VF/AA mutant. Could the authors test ionic current modulation in this mutant at higher Yoda2 concentrations?

The ability of Yoda to modulate stretch currents, but not poking currents, in the beam mutant S1330N/A1479K is quite intriguing and would merit further discussion.

Minor points:

It is important to acknowledge that the grid-based DoGSite3 and AutoDock Vina do not account for the membrane environment and rely solely on static protein conformations. As a result, they are well-known to generate false-positive binding sites and binding poses.

I could not find the supplementary Movie S1.

(Remarks on code availability)

Reviewer #2

(Remarks to the Author)

The entire review can be sent to the authors. See comments.

(Remarks on code availability)

Reviewer #3

(Remarks to the Author)

(Remarks on code availability)

Reviewer #4

(Remarks to the Author)

(Remarks on code availability)

Version 1:

Reviewer comments:

Reviewer #1

(Remarks to the Author)

The authors have adequately addressed my concerns. Congratulations for such a nice work!

(Remarks on code availability)

Reviewer #2

(Remarks to the Author)

The review comments were addressed satisfactorily, I do not have anything further to add.

(Remarks on code availability)

Reviewer #3

(Remarks to the Author)

(Remarks on code availability)

Reviewer #4

(Remarks to the Author)

(Remarks on code availability)

Reviewer #1 (Remarks to the Author):

The authors present robust mutational evidence that two well-known effects of Yoda1 on Piezo1, i.e. spontaneous calcium influx in resting cells and a modulation of mechanically-activated currents, are mediated by separate molecular allosteric mechanisms, likely from the ability of Yoda1 to bind at two distinct sites located within the same inter-repeat binding region of the mechanosensory domain. MINFLUX experiments further reveal that the ability of Yoda1 to spontaneously open the channel correlates with its ability to increase the degree of flattening of this domain, a conformational change known to promote channel opening. The author's molecular interpretation makes sense: Yoda1 binds in the upper site when the channel is mostly closed in resting cells, inducing a partial flattening of the channel and increasing its open probability. The presence of a mechanical stimulus may prompt Piezo to flatten to a higher degree, allowing Yoda to occupy a deeper site and modulate biophysical properties of Piezo currents through a yet unknown mechanism.

We thank the reviewer for her/his overall positive assessment of our manuscript and his/her constructive comments below, which have all been addressed in the revised version of the manuscript.

Major points:

The authors claim that the presence of a secondary, deeper Yoda binding site proposed by Jiang et al. in 2023 has never been tested experimentally (lines 74,136), was exclusively based on in silico data (line 136) and remains speculative (line 287). This is not true. Jiang et al. virtually docked a virtual compound library onto this deeper binding site in their PIEZO1 open state model, which led them to discover two novel activators after experimentally screening ~150 candidates (given the fact that Yoda1 and Jedi activators were randomly discovered from blindly screening millions of molecules, the odds of randomly identifying two unrelated chemical activators from ~150 compounds are close to zero). One might argue that this experimental evidence is indirect (which it is), but the mutagenesis approach used by the authors and by Botello-Smith et al. to probe the Yoda binding site(s) is also indirect, as mutations can have allosteric effects on a remote binding site (as pointed out by the authors lines 319-321). Hence, the two-sites model, first proposed and experimentally tested by Jiang et al., represents the most up-to-date tested model for Yoda activation. The authors ought to introduce this model as early as in the abstract and rearrange their narrative throughout their manuscript in order to properly contextualize their work with the current literature. For example, the subtitle "Computational modelling reveals an alternative Yoda1 binding site that becomes accessible in the flattened state" suggests that the authors have identified a new Yoda1 binding site, while this should be viewed as an independent validation of the Jiang et al. study.

We apologize if our manuscript has left the impression that we did not acknowledge the previous work by Jiang and colleagues enough. This was not our intention. We are well aware that Jiang et al. had already proposed a two-site mode of action and that the discovery of the two novel activators (CMPD15 and CMPD64) based on their in-silico model strongly supports this claim, which is why we had referred to this impressive study throughout the entire manuscript (introduction line 70-73, main text line 132-135, etc.).

We have made even more effort to emphasize the relevance of Jiang et al. for our study in the revised version of the manuscript. Thus, we are now mentioning the two-site mode of action model in the abstract – as suggested by the reviewer – and we have rephrased all statements in the text that insinuated that Jiang and colleagues have not tested the model experimentally.

The reviewer also asked us to rearrange the narrative throughout the manuscript. Although we did not change the order of figures, we paid more attention to properly contextualize our data with previous work in the field. We hope that our revision of the text satisfies the reviewer, with regards to putting our findings into the context of the current literature such that he/she can now recommend publication of our study without reservation.

The authors show that the inability of the 2094W mutant to elicit a calcium influx in the presence of Yoda is likely caused by a loss of binding affinity, as evidenced by an increase of EC50 by two orders of magnitude with no dramatic change in the maximal amplitude of fluorescence signals (Fig 3B). I

wonder whether a similar loss of affinity also explains the lack of Yoda modulation in the VF/AA mutant. Could the authors test ionic current modulation in this mutant at higher Yoda2 concentrations?

We agree that this is an important point to demonstrate that the VF/AA mutation solely affects the Yoda binding affinity and not possible allosteric coupling mechanisms. In fact, we had already shown that 300 nM Yoda2 causes a reduction of the P50 of the VF/AA PIEZO1 mutant (see line 230 of original manuscript and Supplementary Fig.3B). To fully resolve this issue, we have now also examined the effect of additional and even higher Yoda2 concentrations (i.e. 1 μ M and 3 μ M), which also significantly lower the P50 of the PIEZO1 VF/AA mutant. We hope these additional data resolve this issue.

The ability of Yoda to modulate stretch currents, but not poking currents, in the beam mutant S1330N/A1479K is quite intriguing and would merit further discussion.

We agree that this is an interesting observation and had already thought a lot about a possible mechanistic explanation. However, since all attempts to explain the effect are highly speculative and since the observation is not really relevant for the key message of the paper, we eventually decided not to discuss this point in the manuscript.

Since the reviewer, suggests that it might be worthwhile, we have now added a few lines to the results section that focus on this point (line 142-146 of revised paper). We hope that suffices and we also hope the reviewer appreciates that we didn't add more text as this would have interrupted the flow of the manuscript.

Minor points:

It is important to acknowledge that the grid-based DoGSite3 and AutoDock Vina do not account for the membrane environment and rely solely on static protein conformations. As a result, they are well-known to generate false-positive binding sites and binding poses.

We thank the reviewer for pointing this out. We are well-aware of this limitation, which is why we had pointed out (line 187 and line 343 of original manuscript) that the in-silico approach used by Botello-Smith et al. and Jiang et al. is superior, as it also considers the presence of water, lipids and ions. To ensure that the prospective readers of our manuscript do not overlook these statements, we have emphasized them in the revised version by adding the above statement of the reviewer to the discussion (line 358-361 of revised manuscript)

We hope these changes satisfy the reviewer and resolve this issue.

I could not find the supplementary Movie S1.

We apologize that the reviewer had to waste his/her time searching for the movie. There is no Movie S1. We had originally created a movie that showed the transition of Yoda from binding site I to site II using the morphing function of Pymol. While the movie nicely illustrated the two-site induced fit-like mechanism and the transition of Yoda from one site to the other, we felt that the trajectory of Yoda that was shown in the movie was too speculative, which is why we last-minute decided not to include it in the submission.

Reviewer #2 (Remarks to the Author):

Title: 3D-MINFLUX nanoscopy reveals distinct allosteric mechanisms for activation and 2 modulation of PIEZO1 by Yoda1

Review

The mechanically-gated ion channel protein PIEZO1 plays a key role in mechanotransduction across various important physiological processes. Yoda1 is a selective agonist for PIEZO1 and is widely used to study PIEZO1 dependent mechanotransduction. Despite its widespread use, there are still gaps in our understanding of the exact mechanism behind Yoda1 binding and modulation of PIEZO1. In this paper, Verkest and colleagues try to fill in this gap by studying the role of Yoda1 in greater detail using a variety of experimental and computational techniques. Using GcaMP8 calcium imaging and patch-clamp electrophysiology, the authors first corroborate previous results that demonstrated that replacing A2091 with a tryptophan (A2091W) in PIEZO1 suppressed Yoda1 mediated activation.

However, they find that the modulation of mechanically evoked PIEZO1.A2094W currents by Yoda1 was not affected, hinting towards the possibility that a separate binding site may exist for Yoda1-dependent modulation. A further computational analysis using DoGSite3-based pocket analysis and docking pose prediction, combined with previous molecular dynamics simulation results strongly suggest separate binding sites in the flattened conformation of PIEZO1, specifically F1715 and V1714. Experimental results using calcium imaging revealed that the activation of PIEZO1.V1714A_F1715A mutant due to Yoda2 (a more soluble and potent analog of Yoda1) was not affected. However, using patch clamp electrophysiology, it was revealed that the Yoda2-dependent modulation of mechanically evoked PIEZO1.V1714A_F1715A was significantly affected compared to PIEZO1 and PIEZO1.A2094W mutant. This provides strong evidence that V1714 and F1715 binding site plays a critical role only in Yoda-induced modulation but not activation. To further study the flattening of PIEZO1 and its two mutants, the authors employ 3D MINFLUX nanoscopy, a powerful technique that provides nanometer resolution and allows visualization of the channel in great detail. The authors attach ALFA-tags at the distal ends of the PIEZO1 blades and further label the tags with small anti-ALFA nanobody and a DNA that acts as a docking strand for DNA-PAINT. Using an Atto655-conjugated imager strand that can transiently bind and unbind to the docking strand, single fluorophores can be isolated and localized in 3D using MINFLUX. The linkage error, i.e. variation in fluorophore position relative to the actual labeling site on the protein (H86), with this labeling scheme is very small (~2.5 nm). This further helps in improving the interblade distance measurement accuracy. The authors selectively and rigorously identify trimers that represent triple-labeled proteins and compare the mean interblade distance between PIEZO1, PIEZO1.A2094W, and PIEZO1.V1714A_F1715A upon Yoda1 treatment. Their measurements reveal that the mean interblade distance increases for PIEZO1 and PIEZO1.V1714A_F1715A indicating Yoda1-mediated flattening, whereas it did not change for PIEZO1.A2094W. Combined results of the paper indicate that there are two separate binding sites for Yoda-mediated activation and modulation of PIEZO1. Based on these results, the authors propose a two-site induced-fit-like mode of action for Yoda1. Overall, the paper is very well written, the flow and logic are clear. It combines various powerful experimental and computational techniques to provide novel insights into the mode of action of Yoda1, and the discussion section helps bring together the different results to provide a clear picture. The limitations of the study are also explicitly mentioned, for example the uncertainty about whether A2094 and F1715 are the sole binding sites exclusively responsible for the activation and modulation by Yoda1. The results of the study will certainly prove important in guiding the development of novel drugs that can selectively act as agonists and modulators.

We thank the reviewer for her/his overall positive assessment of our manuscript and his/her constructive comments below, which have all been addressed in the revised version of the manuscript.

Following are two concerns, which if addressed, would help improve the quality of the paper.

1. Sample Size and Variability in MINFLUX results:

The number of identified trimers in MINFLUX results for each case are 110 or less, with the lowest being 59 for PIEZO1.V1714A_F1715A + Yoda1 case. While the statistical analysis using unpaired t-test shows low P-values, given the relatively large spread of the interblade distances, it could be sensitive to the variation in sample size. It would be good if the authors can compare the numbers at similar sample size, especially for PIEZO1.V1714A_F1715A where control (N=87) and +Yoda1 (N=59) have >30% variation in sample size. Additionally, it would also be useful to mention the number of different cultures and cells/culture used for obtaining the MINFLUX data for each case, especially given the small and variable field of view used in the MINFLUX measurement (3x3, 5x5, and up to 8x8 μm).

We thank the reviewer for pointing out that differences in sample size could potentially confound the results. To resolve this issue and to strengthen our claims we have increased the N-numbers for the experimental conditions that were undersampled in the original version. Moreover, as requested by the reviewer, we now provide detailed information about the number of cultures that were tested.

Regarding the varying field of view sizes in our experiments, we would like to note that this has logistic reasons. MINFLUX scans are relatively slow: a scan of a small ROIs (e.g. 3 x 3 μm) takes app. 2-3 hours, whereas a scan of a large ROI (8 x 8 μm) requires an overnight imaging session. Since the microscope is located in a core facility, we only have limited access and have to take the imaging slots we get. Accordingly, when we got an overnight slot, we imaged large regions, whereas we imaged multiple smaller regions in shorter imaging sessions during the day. We hope the reviewer appreciates that access to such a high-end state-of-the-art microscope is limited and that we had to adjust our experiments to the availability of the device.

2. Detailed MINFLUX questions: In lines 267 – 270, The variation in the change in interblade distance is within limits of linkage error. The authors need to address this.

To explain and clarify this point, it is very important to note that the variability of the interblade distances under control conditions does **NOT** reflect a possible lack of precision of MINFLUX but in fact shows that PIEZO1 adopts various conformational states in its native environment. In another MINFLUX study from our lab, which we refer to in line 267 (ref 29) and which is now in press at Science Advances, we had examined this phenomenon in great detail by imaging more than 500 PIEZO1 channels. We had demonstrated that PIEZO1 adopts various conformational states including the fully curved state with 0, 1, 2 or 3 ‘handshakes’ (the concept of handshaking of the peripheral blades was recently introduced by another study; PMID:40512861), the intermediate flattened state and the fully flattened state. Our findings are also strongly supported by the recent work of Mazal and colleagues (PMID: 40834076), who also found multiple conformational states of PIEZO1 in its native environment using another cell type and a different imaging approach – i.e. single-particle cryo-light microscopy. Accordingly, Yoda1 (and other drugs and perturbations that alter PIEZO1 sensitivity; see our other study, ref 29) cannot substantially change the maximum and the minimum of the interblade distance distribution, because the max. interblade distance cannot become any bigger than the one of the fully flattened state and not any smaller than in the curved state with 3 handshakes. Hence, Yoda1 solely changes the proportion of PIEZO1 in the various conformational states, which is why the change in the overall mean is rather small.

We hope this explanation resolves this issue. We have also added a similar explanation to the end of the relevant section of the result section of the revised manuscript to ensure that the prospective readers of our paper will be aware of the inherent conformational variability of PIEZO1 in its native environment.

[Figure Redacted]

3. In lines 544 – 546, the authors describe the DBSCAN clustering criteria they used to analyze the MINFLUX data. Were the clustering parameters optimized? Were multiple different combinations (e.g. 3 points and 7 nm, 4 points and 10 nm etc.) attempted? At what threshold did these permutations give statistically, significantly different results? This needs to be addressed.

I think there is a misunderstanding regarding the purpose for which we used the DBSCAN algorithm, which we are happy to clarify:

We used DNA-PAINT to achieve blinking-like fluorescence of our target structure – i.e. the ALFA-tag single domain nanobody. This means that a single anti-ALFA nanobody (which is permanently bound to the ALFA-tag) only transiently becomes fluorescent and thus detectable by MINFLUX when a DNA-PAINT-fluorophore-conjugated imager strand binds. During this binding event, which lasts app. 100 ms, the MINFLUX scan procedure determines the position of the fluorophore multiple times (this series of fluorophore localisation determinations is called a ‘trace’). Once the DNA-PAINT imager strand unbinds from the ALFA-tag nanobody, the detection procedure stops and the MINFLUX laser continues to scan the rest of the ROI until it finds another fluorophore bound to the sample.

Thus, in order to detect a fluorophore the MINFLUX laser needs to be in the exact same position as the target bound fluorophore at the exact same time, which, considering the relatively short duration and low frequency of the binding events (app. 100 ms), is very unlikely. Accordingly, it is even more unlikely that the same target is detected twice during an image acquisition (e.g. once at minute 1 and then again after 30 minutes). Nevertheless, double detections of the same target DO occur, such that one gets two traces for the same ALFA-tag. Since the position estimates of the first and second detection may differ by a few nanometers due to the inherent MINFLUX localisation error, sample drift and other factors, we considered two or more traces (series of multiple localisations, see above) with localisation means that were less than 8 nm apart, as a single trace as they are likely originating from the same ALFA-tag.

It is important to note that such double and triple detections are extremely rare (less than 10% are detected twice and less than 1% detected three times, etc., see figure below), but we nevertheless considered them to avoid that they confound our results. This is why we used the DBSCAN algorithm to detect two or more trace means that are located within 8 nm (see line 544-549 in original manuscript). Hence, the one and only goal of the DBSCAN here is to detect **TWO** or more traces with localisation means within 8 nm and thus increasing ‘minPoints’ above 2 is not necessary. What would be an option is to vary ‘epsilon’ smaller and greater than 8 nm. To resolve this issue we have done this and have analysed our data with more and less stringent epsilon values (6 nm, 7 nm, 8 nm, 9 nm). None of these epsilon values, however, changed the results as can be seen in the figure below.

We hope this explanation and the additional analysis provided here resolve this issue.

4.

there was no text here – I assume '4.' was a typo.

5. Line 562 has typos. (PDB, and PBD).

We have corrected these typos to resolve this issue.

6.

there was no text here – I assume '6.' was a typo.

Point-by-point response to the reviewers comments

Reviewer #1 (Remarks to the Author):

The authors have adequately addressed my concerns. Congratulations for such a nice work!
Thanks for the positive feedback

Reviewer #2 (Remarks to the Author):

The review comments were addressed satisfactorily, I do not have anything further to add.
Thanks for the positive feedback

Reviewer #3 (Remarks to the Author):

I co-reviewed this manuscript with one of the reviewers who provided the listed reports. This is part of the Nature Communications initiative to facilitate training in peer review and to provide appropriate recognition for Early Career Researchers who co-review manuscripts.
Thanks for the positive feedback

Reviewer #4 (Remarks to the Author):

I co-reviewed this manuscript with one of the reviewers who provided the listed reports. This is part of the Nature Communications initiative to facilitate training in peer review and to provide appropriate recognition for Early Career Researchers who co-review manuscripts.
Thanks for the positive feedback

Title: 3D-MINFLUX nanoscopy reveals distinct allosteric mechanisms for activation and modulation of PIEZO1 by Yoda1

Review

The mechanically-gated ion channel protein PIEZO1 plays a key role in mechanotransduction across various important physiological processes. Yoda1 is a selective agonist for PIEZO1 and is widely used to study PIEZO1 dependent mechanotransduction. Despite its widespread use, there are still gaps in our understanding of the exact mechanism behind Yoda1 binding and modulation of PIEZO1. In this paper, Verkest and colleagues try to fill in this gap by studying the role of Yoda1 in greater detail using a variety of experimental and computational techniques.

Using GcaMP8 calcium imaging and patch-clamp electrophysiology, the authors first corroborate previous results that demonstrated that replacing A2091 with a tryptophan (A2091W) in PIEZO1 suppressed Yoda1 mediated activation. However, they find that the modulation of mechanically evoked PIEZO1.A2094W currents by Yoda1 was not affected, hinting towards the possibility that a separate binding site may exist for Yoda1-dependent modulation.

A further computational analysis using DoGSite3-based pocket analysis and docking pose prediction, combined with previous molecular dynamics simulation results strongly suggest separate binding sites in the flattened conformation of PIEZO1, specifically F1715 and V1714. Experimental results using calcium imaging revealed that the activation of PIEZO1.V1714A_F1715A mutant due to Yoda2 (a more soluble and potent analog of Yoda1) was not affected. However, using patch clamp electrophysiology, it was revealed that the Yoda2-dependent modulation of mechanically evoked PIEZO1.V1714A_F1715A was significantly affected compared to PIEZO1 and PIEZO1.A2094W mutant. This provides strong evidence that V1714 and F1715 binding site plays a critical role only in Yoda-induced modulation but not activation.

To further study the flattening of PIEZO1 and its two mutants, the authors employ 3D MINFLUX nanoscopy, a powerful technique that provides nanometer resolution and allows visualization of the channel in great detail. The authors attach ALFA-tags at the distal ends of the PIEZO1 blades and further label the tags with small anti-ALFA nanobody and a DNA that acts as a docking strand for DNA-PAINT. Using an Atto655-conjugated imager strand that can transiently bind and unbind to the docking strand, single fluorophores can be isolated and localized in 3D using MINFLUX. The linkage error, i.e. variation in fluorophore position relative to the actual labeling site on the protein (H86), with this labeling scheme is very small (~2.5 nm). This further helps in improving the interblade distance measurement accuracy. The authors selectively and rigorously identify trimers that represent triple-labeled proteins and compare the mean

interblade distance between PIEZO1, PIEZO1.A2094W, and PIEZO1.V1714A_F1715A upon Yoda1 treatment. Their measurements reveal that the mean interblade distance increases for PIEZO1 and PIEZO1.V1714A_F1715A indicating Yoda1-mediated flattening, whereas it did not change for PIEZO1.A2094W.

Combined results of the paper indicate that there are two separate binding sites for Yoda-mediated activation and modulation of PIEZO1. Based on these results, the authors propose a two-site induced-fit-like mode of action for Yoda1.

Overall, the paper is very well written, the flow and logic are clear. It combines various powerful experimental and computational techniques to provide novel insights into the mode of action of Yoda1, and the discussion section helps bring together the different results to provide a clear picture. The limitations of the study are also explicitly mentioned, for example the uncertainty about whether A2094 and F1715 are the sole binding sites exclusively responsible for the activation and modulation by Yoda1. The results of the study will certainly prove important in guiding the development of novel drugs that can selectively act as agonists and modulators.

Following are two concerns, which if addressed, would help improve the quality of the paper.

1. Sample Size and Variability in MINFLUX results:

The number of identified trimers in MINFLUX results for each case are 110 or less, with the lowest being 59 for PIEZO1.V1714A_F1715A + Yoda1 case. While the statistical analysis using unpaired t-test shows low P-values, given the relatively large spread of the interblade distances, it could be sensitive to the variation in sample size. It would be good if the authors can compare the numbers at similar sample size, especially for PIEZO1.V1714A_F1715A where control (N=87) and +Yoda1 (N=59) have >30% variation in sample size.

Additionally, it would also be useful to mention the number of different cultures and cells/culture used for obtaining the MINFLUX data for each case, especially given the small and variable field of view used in the MINFLUX measurement (3x3, 5x5, and up to 8x8 μm).

2. Detailed MINFLUX questions: In lines 267 – 270, The variation in the change in interblade distance is within limits of linkage error. The authors need to address this.
3. In lines 544 – 546, the authors describe the DBSCAN clustering criteria they used to analyze the MINFLUX data. Were the clustering parameters optimized? Were multiple different combinations (e.g. 3 points and 7 nm, 4 points and 10 nm etc.) attempted? At what threshold did these permutations give statistically, significantly different results? This needs to be addressed.

4.

5. Line 562 has typos. (PDB, and PBD).
- 6.